# Bioactivity and Bioaccessibility of Bioactive Compounds in Gastrointestinal Digestion of Tomato Bagasse Extracts

**DOI:** 10.3390/foods11071064

**Published:** 2022-04-06

**Authors:** Marta Coelho, Carla Oliveira, Ezequiel R. Coscueta, João Fernandes, Ricardo N. Pereira, José A. Teixeira, António Sebastião Rodrigues, Manuela E. Pintado

**Affiliations:** 1CBQF—Centro de Biotecnologia e Química Fina-Laboratório Associado, Escola Superior de Biotecnologia, Universidade Católica Portuguesa, Rua Diogo Botelho 1327, 4169-005 Porto, Portugal; mcoelho@ucp.pt (M.C.); coliveira@ucp.pt (C.O.); ecoscueta@ucp.pt (E.R.C.); jcfernandes@ucp.pt (J.F.); 2LABBELS—Associate Laboratory-CEB-Centre of Biological Engineering, University of Minho, 4710-057 Braga, Portugal; rpereira@deb.uminho.pt (R.N.P.); jateixeira@deb.uminho.pt (J.A.T.); 3Centre for Toxicogenomics and Human Health, Genetics, Oncology and Human Toxicology, NOVA Medical School, Faculdade de Ciências Médicas, Universidade Nova de Lisboa, 1169-056 Lisbon, Portugal; sebastiao.rodrigues@nms.unl.pt

**Keywords:** carotenoids, phenolic compounds, *Lycopersicon esculentum*, ohmic heating, bioaccessibility, bioactivity

## Abstract

A nutrient-rich diet is a key to improving the chemical signals, such as antioxidants, which modulate pathogens’ resistance in the gut and prevent diseases. A current industrial problem is the generation of undervalued by-products, such as tomato bagasse, which are rich in bioactive compounds and of commercial interest (carotenoids and phenolic compounds). This work analyzed the effect of gastrointestinal digestion on the bioactivity and bioaccessibility of carotenoids and phenolic compounds from tomato bagasse extracts. Thus, the extraction by ohmic heating (OH) technology was compared with conventional (organic solvents). The results showed that the main phenolic compounds identified by UPLC-qTOF-MS were p-coumaric acid, naringenin, and luteolin. A higher recovery index for total phenolic compounds throughout the gastrointestinal digestion was observed for OH while for carotenoids, a strong reduction after stomach conditions was observed for both extracts. Furthermore, colon-available fraction exhibited a prebiotic effect upon different *Bifidobacterium* and *Lactobacillus*, but a strain-dependent and more accentuated effect on OH. Thus, the extraction technology highly influenced bioaccessibility, with OH demonstrating a positive impact on the recovery of bioactive compounds and related health benefits, such as antioxidant, anti-hypertensive, prebiotic, and anti-inflammatory properties. Of these properties, the last is demonstrated here for the first time.

## 1. Introduction

Epidemiological studies and associated meta-analyses strongly suggest that long-term eating of fruits and vegetables is key to helping increase immune well-being and prevent diseases [1]. Currently, and especially in this period of COVID-19, food due to its bioactive compounds may create a good health state preventing disease through the improvement of immunity, preventing transmission, and minimizing the effect of the virus at its initial stages [1].

Bioactive compounds need to undergo enzymatic hydrolysis in the digestive tract or be metabolized by the bowel microbiota to be absorbed [2]. Their bioavailability and bioaccessibility could depend on the capacity of extraction methods to improve their recovery [2].

Also, the ever-growing demand to recover bioactive compounds from by-products encourages a constant search for accessible extraction methods [1]. A few studies have applied green technologies as a new extraction method. An example is ohmic heating (OH), used in the food industry to pasteurize [3]. The OH ohmic heating technology uses electrodes to reach high temperatures in seconds. These electrodes are in contact with the desired matrix (for example, food). An electric current is generated that passes through this sample, generating an electric field. It is possible, for example, to pasteurize extract compounds without using other toxic solvents. Therefore, green, sustainable, low-cost technology is an excellent alternative to conventional methods (CONV) [4,5,6].

Barba et al., (2015) applied a pulsed electric field, which uses electric voltage pulses, being solvent-free as well. The main reasons for the increased bioaccessibility verified in this study are unknown. However, it may be due to the electroporation phenomenon causing electrical breakdown and allowing the perforation of the cytoplasmic membrane promoting leakage of cell content, improvement of bioactive compounds, and probably solubilization and digestion [7]. Also, the use of extracts rich in bioactive compounds does not mean that all of them, when ingested, can pass into the bloodstream and cause beneficial health effects [2,8].

The tomato paste industry generates significant amounts of bagasse, a valuable by-product rich in health-promoting compounds including polyphenols and carotenoids [9]. The main phenolic compounds present in this by-product are gallic acid, naringenin, chlorogenic acid, and rutin [10]. Lycopene and β-carotene are the most carotenoids present in this by-product [11]. 

Some authors have studied the bioaccessibility of tomato fruit [9,12]. Nevertheless, there is no straightforward correlation between process-induced matrix delay and bioactive compounds extractability bioaccessibility and health effects. Indeed, no studies have compared the bioaccessibility impact of ‘compounds’ rich extracts obtained from new extraction technologies, such as ohmic heating (OH), with CONV extractions methods.

Therefore, the purpose of this study was to evaluate the in vitro gastrointestinal digestion (GID) on the stability, recovery, and biological properties of carotenoids and phenolic compounds isolated by OH technology and conventional method (organic solvents) from tomato by-products. A static in vitro GID method was monitored to simulate the digestion (categorized into salivary, gastric and intestinal) to determine (1) whether the mouth, stomach or intestine compartment plays a role in the bioaccessibility of both groups of compounds; and (2) how the extraction conditions (OH and CONV extract) affect the bioaccessibility and related biological properties of extracted bioactive compounds throughout the GID. 

## 2. Materials and Methods

### 2.1. Chemicals

The 2,20-azo-bis-(2-methylpropionamidine)-dihydrochloride (AAPH), fluorescein, 2,2-azinobis-3-ethylbenzothiazoline-6-sulphonic acid (ABTS diammonium salt), potassium sorbate, sodium carbonate, ethylenediaminetetraacetic acid (EDTA), sodium sulfite, and sodium lauryl sulfate, lipopolysaccharide from O111:B4, and *Esherichia coli* were purchased from Sigma-Aldrich (Sintra, Portugal). Hexane, ethanol, Folin–Ciocalteu’s reagent, and potassium persulfate were purchased from Merck (Algés, Portugal). The enzymes α-amylase, pepsin, from porcine stomach mucosa pepsin A 250 UI/mg, pancreatin from porcine pancreas, and bile salts (Sigma-Aldrich, Taufkirchen, Germany). The fetal bovine serum, RPMI 1640, glutamine, Pen/Strep (Gibco, New York, NY, USA), ELISAs from Biolegend, CA, USA. Standards of ascorbic acid, Trolox, gallic acid, rutin, *p*-coumaric, and 4-hydroxybenzoic acid, were purchased from Sigma-Aldrich (Sintra, Portugal), while kaempferol, β-carotene, lycopene, zeaxanthin, and lutein (Extrasynthese, Lyon, France) were purchased from Extrasynthese (Lyon, France). 

### 2.2. Samples

The study was performed with tomato bagasse liquid extracts obtained in previous work (Coelho et al., 2019). Briefly, both phenolic and carotenoids extracts were performed by using two methods, ohmic (OH) and conventional (CONV) extraction, applied 20 mL of ethanol (70%) and hexane to 2 g of tomato (H1015 Heinz seeds) bagasse (seeds, skins, and pulp) provided by Centro Competencias Tomate Indústria from the centre of Portugal An experimental design was also applied to obtain OH extracts with yields closed to the obtained by CONV method. After, the optimized extracts obtained were 70% ethanol, 70 °C during 15 min of extraction for phenolic compounds, and 70% ethanol, 55 °C during 15 min for carotenoid extracts [13]. The liquid fractions (LF-OH and LF-CONV) obtained were submitted used in the following analysis. 

### 2.3. In Vitro GID

Simulated complete digestion of the LF (LF-OH and LF-CONV) was performed according to the method described by Coelho et al., (2021). This procedure mainly comprised sequential phases simulating different conditions along the GID. The mouth step simulation was performed with 2 g of each sample diluted in 2 mL of PBS solution and adding α-amylase from human saliva (100 U/mL in 1 mM aqueous CaCl2), after incubating for 5 min at 37 °C under agitation (200 rpm). After, the gastric stomach phase was simulated with the pepsin solution (12.5 mg/mL in HCl 1 M) at a 0.05 mL/mL of sample ratio, at pH 2.3 following incubation for 2 h at 37 °C at 130 rpm orbital agitation. Finally, the small intestine phase was simulated with the addition of 20 mL of a solution composed of 1.98 mg of pancreatic and bile extract solution at pH 7 and incubated for 3 h at 37 °C with 45 rpm. After the samples were placed into cellulose membrane dialysis tubing and dialyzed against a total of 2 L of PBS for 24 h (changing the PBS twice) to simulate the colon (non-absorbable sample) and the liquid outside the membrane means basolateral part, during 24 h at 37 °C with 45 rpm. After the gastrointestinal simulation, the basolateral fraction was freeze-dried for subsequent analysis.

During the gastrointestinal simulation, samples were collected (2 mL) in each step: mouth; stomach, small intestine, colon (IN) and basolateral (OUT) fraction to analyze total phenolic compounds, total carotenoids, and qualitative and quantitative profiles of both phenols and carotenoids by HPLC and UPLC-q-TOF MS.

After gut digestion, the digested OH and CONV extracts and their bioactive properties were measured (antioxidant, prebiotic activities and anti-hypertensive activities). All analyses were performed in triplicate.

### 2.4. Recovery and Bioaccessibility Indexes of Polyphenolic and Carotenoids Compounds throughout In Vitro GID Digestion

The recovery percentage determines the principal compound amount during each step of GID following the equation:Recovery index (%) = (BCDF/BCTF) × 100,

BCDF represents the digested bioactive content (mg), and BCTF is the bioactive (mg) quantified in the test matrix.

Bioaccessibility is defined as the percentage of the bioactive compounds (BCS) solubilized after intestinal dialysis step; this index defines the proportion of the bioactive compound that could become available for absorption into the blood system:Bioaccessibility index (%) = (BCS/BCDFE) × 100
where: BCS is the bioactive content (mg) in the digested sample after the dialysis step (OUT) and BCDFE is the bioactive content (mg) in the digested sample after the intestinal step (IN + OUT)—end of digestion. Whereas OUT represents the fraction passed through the dialysis membrane to the liquid involved after digestion (simulation of the basolateral fraction).

### 2.5. Analysis of Gastrointestinal Fractions

#### 2.5.1. Total Phenolic Content (TPC)

The total content of phenolic compounds present in the extracts was evaluated through the Folin-Ciocalteu spectrophotometric method described by Coelho et al., (2021). Both OH and CONV extracts were measured using the spectrometric method and read in microplate read (Sunrise Tecan, Grödig, Austria) at 750 nm. The TPC content was expressed in mg gallic acid equivalent per g dry weight material (mg GAE/g). All of the analyses were performed in triplicated, and the standard deviation was calculated. 

#### 2.5.2. Total Carotenoids Content (TCC)

The TCC present in the OH and CONV extracts were measured through the spectrophotometric method [13] and read at 454 and 536 nm [14] using a microplate reader (Sunrise Tecan, Grödig, Austria). The content of total carotenoids was expressed as milligram β-carotene equivalent per dry weight material (g_β-carotene equivalent_/Kg). The analyses were performed in triplicate, and a standard deviation was calculated.

#### 2.5.3. HPLC-Analysis (Phenols and Carotenoids)

Qualitative and quantitative profiles of phenols were carried out according to the method proposed by Oliveira et al., (2015) with slight modifications, as described in Coelho et al., (2021). The phenolic compounds were analyzed in a Waters Liquid Chromatograph (Waters Series 600. Milford, MA, USA) with A C18 guard column (Symmetry^®^ C18) and an Alltech adsorbosil C18 reversed-phase packing column (250 × 4.6 mm i.d. 5 μm particle size and 125 Å pore size) for compounds separation throughout this study. The PDA acquisition wavelength was set in 216–600 nm, analogue output channel A at a wavelength of 280 nm and analogue output channel B at 360 nm, both with a band with 2 nm. A - Acetonitrile (100%) with 0.2% TFA; Solvent B: acetonitrile/water (5:95 *v*/*v*) (Merck pure grade and pure water) with 0.2% TFA (Sigma-Aldrich, Germany). Samples were analyzed in triplicate. Calibration curves of were obtained at a detection wavelength 280 nm (flavan-3-ols), 320 nm (flavonols). Standards solutions rutin, kaempferol, naringenin, luteolin, vanillin, 4-hydroxybenzoic acid, transcinnamic acid, and caffeic acid were prepared to identify and quantify phenolic compounds over the concentration range from 0.10 to 100.00 mg/L and expressed as micrograms per mL of dry weight (DW) biomass of tomato. All calibration curves were linear over the concentration ranges tested, with correlations coefficients of 0.999.

Qualitative and quantitative profiles of carotenoids were analyzed according to the method proposed by Oliveira et al. [15]. It used a HPLC-DAD in a Vydac 201TP54 C-18 column (250 × 4.6 mm), equipped with a C-18 pre-column. The phases were composed of a solvent A with ethyl acetate (Merck pure grade) and solvent B 90:10 acetonitrile:water (Merck pure grade and pure water, 1.0 mL/min flow rate, at room temperature. The UV–vis detector was set between 270 and 550 nm. Individual carotenoids were quantified based on a calibration curve built with pure standards: β-carotene, lycopene, zeaxanthin and lutein (Extrasynthese, Genay Cedex, France) and expressed as milligrams per kilogram of dry weight.

#### 2.5.4. UPLC-qTOF MS Analysis (Phenolic Compounds)

The UPLC-qTOF MS allows an analysis of the complete compound profile and its derivatives, which is not possible by HPLC-DAD. This analysis was carried out according to the method described in Coelho et al., 2021. Briefly, an Ultimate 3000 Dionex UPLC coupled to an ultra-high resolution Qq-time of flight (UHR-QqTOF) mass spectrometer (Impact II, Bruker Daltonics, Germany) was used to analyze the phenolics. Data were acquired in negative mode (50 to 1500 amu), and an auto MS/MS scan mode. The MS parameters were: capillary voltage: 2.5 kV; drying gas temperature: 200 °C; drying gas flow: 8 mL/min; nebulizing gas pressure: 2 bar; collision RF: 300 Vpp; transfer time: 120 µs and pre-pulse storage: 4 µs. Mass calibration was performed by the external injection of a sodium formate solution. The accurate mass measured was within 5mDa of the assigned elemental composition, and mSigma values of <20 provided confirmation.

### 2.6. Bioactivities

#### 2.6.1. Antioxidant Activity

Both antioxidant activity (AA) and total phenolic compounds content were measured in samples before and after in vitro digestion. Each sample’s in vitro antioxidant activities were directly evaluated in the lyophilized powdered fractions using the 2,2′-azinobis-(3-ethylbenzothiazoline-6-sulfonic acid radical cation) (ABTS^+^) and oxygen radical absorbance capacity (ORAC) methods, as described by Coelho et al., (2021). All of the analyses were performed in triplicated. ABTS measures the antioxidant activity. Briefly, samples (10 μL) were added to a colored solution of ABTS^•+^, with an optical density (OD) measured at 734 nm and adjusted to 700 ± 0.020 in a spectrophotometric microplate reader (Sunrise Tecan, Grödig, Austria). After 6 min of reaction, the final OD was read was expressed in g ascorbic acid equivalent per Kg DW. In the ORAC method, the extracts were dissolved with phosphate buffer (pH 7.4), and a standard curve of Trolox was performed ranging from 0–90 mg/L. At the analysis time, 70 nM fluorescein and 14 mM AAPH results were made at ORAC buffer. The 96 wells colored microplate was prepared to contain25 µL of blank control (ORAC buffer); standardized, control, or sample and 200 µL of fluorescein solution were added. Then, 50 µL of newly prepared AAPH solution was added. The microplate was incubated for 10 min at 37 °C. The fluorescence readings were carried every 2 min within 104 cycles using the FLUO star OPTIMA plate reader (BMG Labtech, Offenburg, Germany). The wavelength excitation was 485 nm, and the emission was 530 nm. Results were expressed in mol Trolox equivalent per Kg DW, and each sample’s measurements were performed in triplicate [16].

#### 2.6.2. Prebiotic Effect

Based on the most common group of bacteria used to test this property, six commercial probiotic bacteria were selected for the present work, namely *Bifidobacterium animalis* subsp. *lactis* BB-12; *Bifidobacterium longum* BG3; *Bifidobacterium animalis* Bo; *Lactobacillus acidophilus* LH5; *Lactobacillus casei* LC1; and *Lactobacillus rhamnosus* R11.

The effect of tomato bagasse extracts on the target probiotic microorganisms’ growth was evaluated. The solutions prepared using tomato bagasse extract (before and after digestion) and basal media were prepared at 2, 4 and 6% (*w*/*v*) and inoculated, using a 24 h inoculum, at 10% (*v*/*v*). These bacteria were then incubated for 24 h at 37 °C, and Bifidobacterium strains were inoculated under an anaerobic environment. After this period, the cell growth was dictated by plating, using the spread plate method in de Mann, Rogosa, and Sharpe agar (MRS) enhanced with 0.5 g/L of L-cysteine hydrochloride. After 48 h incubation at 37 °C under anaerobiosis, the colonies were enumerated, and the outcomes were plotted as log CFU/mL. All inoculations were performed in triplicate, and the plain inoculated basal media was utilized as a control.

#### 2.6.3. ACE-Inhibitory Activity Assay (iACE)

The ACE-inhibitory activity (iACE) was determined by the fluorimetric assay described by Coscueta, Brassesco and Pintado (2021) [17]. This method is based on the ability of the angiotensin-I converting enzyme to hydro-lyse a specific substrate (o-aminobenzoylglycyl-p-nitrophenylalanyl-proline (Abz–Gly–Phe(NO2)–Pro)), generating the fluorescent product o-aminobenzoylgly-cine (Abz–Gly). iACE of each sample was evaluated in triplicate and expressed as the concentra-tion capable of inhibiting 50% of the enzymatic activity (IC50). To calculate the IC50, non-linear modelling was used, and the results were expressed as μg mL^−1^ to inhibit 50% of the enzymatic activity.

#### 2.6.4. Citocines Inhibition—Anti-Inflammatory Activity

Peripheral blood mononuclear cells (PBMCs), namely lymphocytes and monocytes, were isolated from whole blood obtained from healthy donors of whom informed consent was obtained that their donated blood would be used for scientific purposes [18]. Separation of blood cells was performed using density centrifugation. After isolation, PBMCs were washed twice in cold phosphate-buffered saline (pH = 7.4) containing 3% heat-inactivated fetal bovine serum. The viability of the PBMCs was evaluated using a Neubauer counting chamber using the trypan blue exclusion test.

Tomato absorbed fraction treated by CONV, and OH technology was tested for its pro- or anti-inflammatory properties on PBMCs simulated or not with lipopolysaccharide at 100 ng/mL. Isolated PBMCs were then plated at a density of 1.0 × 106 cells/mL in RPMI 1640, supplemented with 10% heat-inactivated fetal calf serum, 2 mM glutamine and 100 U/mL of Pen/Strep in a humidified atmosphere containing 5% CO_2_ for 24 h. After 24 h incubation, supernatants were harvested by centrifugation and different pro-inflammatory cytokines concentrations (TNF-α and Il-6) were measured by commercially available ELISAs according to the manufacturers’ instructions.

### 2.7. Statistical Analysis

Statistical analysis was conducted using IBM SPSS Statistics v21.0 (IBM, Chicago, IL, USA). The normality of the data’s distribution was evaluated through Shapiro-Wilk’s test. As the data proved to follow a normal distribution, One-way ANOVA, coupled with Tukey’s post hoc test, was used to determine the differences in mean values between Bioactive compounds or bioactivities concentrations and digestion. Pearson’s test assessed the correlation between total phenolic compounds, carotenoids, individual compounds, and bioactivities. Tomato bagasse biomass’ effect on bacterial populations, at each time point. Repeated Measures ANOVA was used to evaluate the effect of Tomato bagasse biomass on the bacterial population over time. Differences were considered significant for *p*-values ≤ 0.05.

## 3. Results and Discussion

### 3.1. Extract Phenolic and Carotenoids Characterization

#### 3.1.1. Phenolic Characterization

The results showed significant differences between the OH and CONV methods for most phytochemicals analyzed (Figure 1). The TPC of OH extract was 2.15 ± 0.049 g/Kg DW while the CONV present higher values 4.23 ± 0.064 g/Kg DW compared with OH. Similar values were found by Coelho et al., 2019, whereas the authors applied an experimental design to optimize the phenols and carotenoid extraction yields with OH. They found TPC values ranging from 0.480 and 2.83 g GAE/Kg DW and concluded that temperature, solvent (ethanol: water ratio) and time are essential to obtain higher extraction yields of bioactive compounds. Besides, this method could be selective (i.e., different conditions allow the extraction of different bioactive compounds such as polyphenols and/or carotenoids compounds). Other studies reported TPC values for peels ranging from 0.44 and 6.10 g GAE/Kg [19]. The discrepancies found in the results indicated that an ethanol-water solution did not recover the major phenolic compounds present in tomato bagasse. These differences may occur since tomato bagasse contains mainly peel and seeds, and the insoluble polyphenols are covalently bound to cell wall components, such as hemicellulose, cellulose, lignin, structural proteins, and pectin. Also, the various extraction processes release the phenolic compounds from the matrix in which they are contained in differently [19].

OH could be a promising technology over the CONV method to reuse all tomato bagasse despite the results. Previous studies significantly influenced bioactive compound recovery by setting temperature, time, and ethanol concentration [5,20]. Furthermore, a synergetic effect between temperature and electric fields caused by OH has been verified, facilitating the cell walls breakdown and allowing polyphenols’ bioaccessibility with higher recoveries in short periods.

Regarding HPLC and Q-TOF analysis, the most abundant phenolic compound in both extracts from tomato by-products is rutin, followed by kaempferol, luteolin and naringenin (Figure 1A). However, the number of phenolic compounds changes with the type of extraction used. Other compounds were also found, such as phenolic acids, p-coumaric acid, caffeic acid, and hydroxycinnamic acid. These discrepancies could be explained by the type of extraction used, which influence the individual compounds [21]. The effect of the extraction process was evidenced in the different parameters evaluated in this study. The OH extracts presented higher rutin content than the CONV samples (*p* < 0.05), while to other phenols, the concentrations between extraction methods were very similar. Rutin is better realized in hydrophilic solutions than hydrophobic, such as the CONV method used, which promote its higher recovery yield with OH application [21].

UPLC-q-TOF MS only identified some compounds. These results express the enormous potential of tomato bagasse in diverse phenolic compounds. Excellent mass accuracies were observed for all molecular ions, presenting differences between experimental *m*/*z* values and calculated *m*/*z* values below 2 mDa. The most representative phenols and derivatives profiles from OH and CONV extracts are present in Table 1 and revealed the similarity of compound profile between OH and CONV, differing in the peak intensity. The OH extract differs in the presence of sucrose, which proves the affinity of hydrophilic compounds in the OH extraction. Also, for the first-time derivatives of kaempferol, such as kaempferol 3-sophorotrioside, and N-acetyl-D-tryptophan, were found and described in the literature. The results showed the presence of flavonol and derivates such as rutin at 609 *m*/*z*, kaempferol 3 -sophorotrioside at 771 *m*/*z*; quercetin-3-*O*-neohesperidoside at 609 *m*/*z* [22]. Naringenin at 271 *m*/*z*, a flavanone and the primary phenolic compound present in tomatoes by-products, was also reported by other authors [22]. Hydroxycinnamic acids were also present, namely *p*-coumaric acid at 163 *m*/*z*. The phloridzinyl glucoside at 597 *m*/*z* was also present and is a dihydrochalcone, a group of polyphenols often forgotten but essential for its biological properties [23,24].

#### 3.1.2. Carotenoids 

Figure 1B, showed carotenoid content in extracts obtained by CONV and OH methods. 

Although chlorophylls and carotenoids present higher values in the CONV relatively to the OH method, the values obtained in OH were reasonable, considering the lipophilic structure of these compounds and the solvent used (70% ethanol). 

This observation is consistent with the results reported by Nguyen and colleagues (2001). These authors studied the thermal isomerization of carotenoids from different tomato varieties. They also attribute the structural lycopene stability relative to β-carotene to its thermodynamic nature due to the chemical structure difference and the intracellular localization of lycopene (chloroplasts). Calvo et al., (2007) studied the influence of ethanol and ethyl acetate and temperature on carotenoid extraction from tomato peel powder. They observed an increase of carotenoids recovery with temperature increase to 50 °C and a decrease when the temperature reached 60 °C, suggesting an oxidative degradation of carotenoids. Besides, they observed better extraction yields for ethanol than ethyl acetate. The type of solvent, the concentration used to extract, and its polarization influence polyphenols and carotenoid extraction.

Regarding CONV and OH methods used to extract bioactive compounds from tomato bagasse, significant differences between methods were only observed with β-carotene. No differences were found between CONV and OH methods for other compounds found. Even though the three compounds (β-carotene, lycopene, and lutein) are structurally similar, they perhaps present different solubilities in the solvents and specific solvency conditions of temperature. Different studies [9,13,25,26] have described that expanding the temperature builds the solvency of the carotenoids. This conduct could explain how the carotenoid concentrations in the different segments of the vegetable tissues are distinct and how lycopene crystallizes inside chromoplasts [5,27]. An explanation is that temperature improves lycopene extraction by expanding the solvent diffusivity (and the capacity to enter into the robust matrix) [5]. This reality hampers lycopene extraction since the chromoplast cell wall is a boundary. Furthermore, the disintegration of a crystallized substance is slower than that of an amorphous one, while the solid:liquid ratio is low the intermolecular forces in the crystal are most potent [26].

As expected, significant differences were found in total carotenoid content extracts (*p* < 0.05) (Figure 2). Although the OH was carried out using 70% ethanol, the total carotenoids recovery was 2.07 ± 0.09 g/kg DW against 1.92 ± 0.06 g/Kg DW of the CONV method that uses organic solvents. Nonetheless, when the carotenoids profile was analyzed, it was interesting to observe that the impact of the system on each compound was different. The CONV extracts present more β-carotene than OH extracts, while the OH extracts present more lycopene with a significant difference (*p* < 0.05). Both compounds have different chemical structures, which may influence their hydrophobicity, solubility, and, consequently, their thermal stability. Lycopene is a linear molecule with a more stable structure, which may form multilayers and aggregates resisting further structural changes, while β-carotene and lutein have unstable β-ionone rings and are not available for molecular self-assembling [9,25,28]. In OH a temperature of 55 °C was applied on tomato bagasse, not enough to degrade lycopene, but enough to break down protein/polysaccharide-lycopene complexes, releasing more lycopene into the solution. Also, ethanol, the solvent used in OH, possibly leads to a better penetration inducing a selective and physical cell disruption, allowing the recovery of lycopene. Thermal treatments cause oxidation and isomerization of β-carotene [13,25,28]. Thus is the first time that OH was proven to preserve lycopene compared with CONV methods.

### 3.2. Recovery and Bioaccessibility Indexes of Polyphenolic and Carotenoids Compounds throughout In Vitro GID

#### 3.2.1. Phenolic Compounds

The TPC and individual phenolic compounds detected during the in vitro GID are presented in Table 2 and Figure 2. The individual compounds were identified by UPLC-qTOF-MS (Table 2) and quantified by HPLC-DAD.

A statistically significant increase (*p* < 0.05) was observed regarding TPC between undigested (extract) and digested fractions of OH extracts. As noted earlier, OH extracts contain fewer total phenolics than those obtained by the CONV method (*p* < 0.05). Following the digestion, the OH samples presented lower TPC than CONV samples in the mouth (*p* < 0.05), nevertheless, the OH extracts allowed the more significant release of compounds than the CONV method (RI 131% and RI 92%, respectively).

These results showed that the oral digestion phase affected tomato extracts and their bioactive compounds differently since the release of polyphenols from the extracts matrix depended on the type of OH and CONV extraction and its bioactive compounds (*p* < 0.05). Thus, the different composition of tomato bagasse extracts present at the mouth, namely the soluble fiber content (data are not shown), which is higher in OH than in CONV, certainly influences the bioaccessibility of polyphenols and the interaction of α-amylase and polyphenolic compounds. Certain bioactive compounds, such as phenolic compounds, could be linked to soluble fibre, and the digestive enzymes could hydrolyze and release the phenolic compounds in ester and glycoside forms present in a higher amount in OH than the CONV extracts. Also, the use of OH during the extraction process could improve the release of compounds since this method uses the electric fields, which could also change the cell wall and the molecular structure and consequently could improve the release of polyphenols towards the matrix [5,7], leading to these differences in the recovery index [29]. Also, the TPC amount of OH extracts may be explained by the higher content of hydroxybenzoic acid, rutin, naringenin caffeic acid, and luteolin compared with CONV samples.

No differences were found for the TPC in the stomach and small intestine for both extraction methods, presenting similar TPC (*p* > 0.05). Nevertheless, the stomach presented a higher impact on the TPC of OH extracts leading to a higher TPC released observed in the small intestine. The TPC recovery in OH extract is higher than in the CONV samples (RI 141% and RI 76%, respectively). The acidic pH in the gastric step promotes the breakage of bonds between bioactive compounds and other extracts components, such as dietary fiber or protein, which release polyphenols into the matrix. Other authors reported the increase of phenolic compounds recovery in the gastric step [2,29,30,31,32].

Regarding the intestinal fraction, the OH also presents a higher percentage recovery of TPC than the CONV samples (RI 33% and RI 23%, respectively). The differences obtained between extracts may be explained by (i) the presence of lutein in OH extracts in this fraction; (ii) due to the interactions between carbohydrates, dietary fibers or proteins [33]; (iii) due to chemical reactions, such as oxidation and polymerizations of bioactive compounds, which lead to the formation of other derivatives, such as chalcones; and (iv) due to the structural changes associated to enzymatic actions, which cause influence in the solubility [2].

In the basolateral fraction, the OH extracts showed lower TPC than the CONV samples (*p* < 0.05), but OH present a recovery index of TPC higher than the CONV extracts (RI 84% and RI 73%, respectively). This difference could be promoted by the presence of caffeic acid in OH extracts at this step. Possibly, one of the reasons for the higher TPC content of CONV is that this process is more aggressive upon other structures present in the extracts that facilitate the increasing availability of phenols and consequently higher absorption capacity towards the bloodstream (here represented by a basolateral fraction) [2]. Also, the recovery index differences could be explained by the type of extraction used and the bioactive compounds present in the extracts [7]. The more polar compounds, such as benzoic acids or hydroxycinnamic acids are not conveniently extracted by CONV methods, which mainly uses organic solvents, and at the same time, the OH uses electric fields, which makes phenols more accessible in the extracted matrix [5,34].

Globally, the highest TPC recovery values along the digestion process were observed in the oral phase and gastric digestion of OH extracts (*p* < 0.05). In the stomach, the TPC value increased for both extracts (*p* < 0.05) and decreased in the intestine phase (*p* < 0.05). The recovery indexes were consistently lower in the case of the CONV extracts. 

As previously mentioned, variations such as pH and acidity in the gastric phase promote the breaking of bonds between bioactive compounds and nutrients, such as fibers, proteins, and carbohydrates. Also, the acidic conditions of the stomach protect the phenols from degradation [2,29].

The TPC decreases in the colon fraction (*p* < 0.05) and their increase in the basolateral fraction (*p* < 0.05) may indicate the absorption of phenolic compounds in the large intestine. This fact is confirmed by the results obtained for individual phenolic compounds, namely, naringenin, rutin, hydroxybenzoic acid and caffeic acid (Table 2). Furthermore, the additional contact time (24 h) of the extract and intestinal fluids (including the splitting enzymes, namely lipolytic, amylolytic, proteolytic) allows the release of polyphenols [7,33]. The same results were obtained with tomato sauce [35] and olive pomace [2].

Regarding the individual polyphenols compounds present in both extracts, the results obtained by HPLC followed the TPC results. High correlation coefficients (r^2^ ≥ 0.73) were obtained with the GID for TPC and individual compounds for both extractions. The individual compound with a higher correlation coefficient in OH extracts is naringenin (r^2^ ≥ 0.91), followed by trans-cinnamic acid (r^2^ ≥ 0.79) and rutin (r^2^ ≥ 0.73). On the other hand, the best correlation of the CONV extract was observed for 4-hydroxybenzoic acid (r^2^ ≥ 0.95). This correlation between individual compounds and TPC is evident for the recovery index in each step. Regarding the naringenin content for OH extracts, it increased in the mouth (RI 107%), followed by an increase in the stomach and small intestine (RI 150% and RI 224%, respectively) and a decrease in the basolateral phase (RI 13%), which represents the bioaccessible blood. The non-absorbed fraction interacting with gut microbiota. For CONV the recovery index of 4-hydroxybenzoic acid increased in the mouth fraction (RI 11%), followed by a decrease in the stomach (RI 1%) and an increase after the gastric step (RI 2.50%) as well as in the intestinal process (RI 8%) with values far from the OH extract.

UPLC-qTOF-MS allow identifying other individual compounds and their metabolites during the GID step. The principal component analysis (PCA), please see Figure 3, was performed to identify a global pattern throughout the GID of tomato extracts under the extraction conditions applied, OH and CONV. According to PCA, the two main components (first present in the horizontal axis and the second the vertical axis) accounted for 77% of the variable in the data assessed, representing a satisfactory analysis. The OH and CONV extracts confirmed their differences in the type of compounds content, and also they are similar in terms of absorption.

Furthermore, applying the PCA and co-expression analysis (fold change analysis), applying the Pearson correlation coefficient allowed to find the correlations between compounds in the dataset, eliminating the biased results. Thus, looking for correlation coefficients, it was possible to understand the main compounds/metabolites, which interfere throughout the GID in OH and CONV samples. The results obtained showed that the main compounds which interfere with gastrointestinal results are namely *p*-coumaric acid (163 *m*/*z*), naringenin (271 *m*/*z*), and 4-coumarate (601 *m*/*z*). The impact of extraction methods in each compound recovery is also presented in Figure 3, with OH affecting the naringenin and chalcone and in the CONV extraction, quercetin (271 *m*/*z*) was the most affected. Analyzing the results following the literature, the OH allows the recovery of more hydrophilic compounds than the CONV method since the former uses apolar solvents [5].

The intestinal phase represents the most significant step in GID, where the main bioactive compounds are absorbed in the epithelium. Thus, it is crucial to define bioaccessibility as the amount of an ingested bioactive compound available for absorption.

The digested fractions obtained after the GID process generate different soluble and insoluble compounds, potentially bioaccessible molecules [36]. 

The results showed that 4-hydroxybenzoic acid is the most bioaccessible compound for OH extract, and also with higher bioaccessibility than when present in the CONV extract (24.32% and 3.27%, respectively), followed by rutin, which presented much lower bioaccessibility than in the CONV extract (10.41% and 78.78%, respectively). Naringenin showed to be only bioaccessible in the OH extract (6%). Thus, the polyphenols’ chemical structure defines their rate and extent of intestinal absorption and the type of metabolites circulating in the blood [29]. These compounds have been reported with beneficial properties such as antioxidant and antidiabetic properties, among other properties [3]. 

#### 3.2.2. Carotenoids

Concerning the total carotenoid content along the GID present in both extracts, OH and CONV (Table 2, Figure 2B), a significant decrease between extracts and mouth was observed for OH extracts (21.25%) and CONV (22.97%). The high sensitivity of carotenoids to external factors may explain the higher carotenoid losses, which led to similar values of 79.35% in OH and 78.42% in CONV extracts after stimulation of enzymatic digestion acidic conditions than in the mouth [37,38].

Moreover, although the OH extract presented similar values to the CONV in terms of total carotenoid content along the digestion process, the recovery index of carotenoids in the basolateral fraction was 5.76 ± 2.43% for OH extract compared to 22.97 ± 3.12% from CONV extract. The more hydrophobic profile of extracts, hydroxyl groups, and type of carotenoids may explain the higher CONV extracts bioaccessibility [39].

After intestinal digestion, the total carotenoid decrease is probably due to interactions with other compounds or changes produced by the hydrolysis of enzymes present in the digestion process and pH changes in each compartment. Besides, carotenoids are lipophilic and have reduced bioaccessibility in the GID due to poor absorption in the gastrointestinal fluids [40].

### 3.3. Bioactive Properties of Liquid Extracts after Simulated Enzymatic GID

#### 3.3.1. Antioxidant Activities of Liquid Digest Extracts throughout In Vitro GID

The results obtained from ABTS and ORAC assays (Figure 4A and Figure 4B, respectively) showed significant differences concerning total AA regarding the two extraction methods and the digested fractions analyzed (*p* < 0.05).

Regarding the AA measured by ABTS, before digestion (extract), OH exhibited lower AA than CONV extracts. These results were almost certainly due to the differences among tomato extracts in terms of compounds, hydrophilicity, solubility and accessibility of their compounds [36].

OH extracts were evaluated with higher AA than the CONV extracts (*p* < 0.05). As the results indicate, OH allowed compounds from the matrix to the gastrointestinal fraction (Table 2) [5,7]. 

After mouth and gastric simulations, the AA of OH samples increased while CONV samples decreased. Both extracts displayed an increase in the total AA following enzymatic GID. After releasing and absorbing bioaccessible antioxidants in the small intestine, the AA exhibited by the compounds that reach the colon was again higher for OH than for CONV. The highest AA was provided by OH extracts in the colon, despite the slightly lower fermentability estimated for this extract (ethanolic water extract, which contains lower dietary fiber) [41]. Also, in general, the digested fractions showed significantly higher AA than their respective undigested extracts. The enzymatic GID phase produced a marked increase in the AA of the OH extracts. The antioxidant effects depend not only on their (polyphenols, carotenoids) concentrations in foods but also on bioaccessibility after ingestion [5,31,42].

Regarding phenolic compounds, metabolism from liquid extracts, which contains lower dietary fiber, practically begins within the lumen of the small digestive system and can pass through the intestine wall into the bloodstream In contrast, post absorption suffers modifications within the liver and other organs. For example, to the flavonoids retention from the small digestive tract occur, the glycosides (sugars) that are linked to the flavonoid skeleton must be the first broken by the enzyme’s activity of small intestine (e.g., β-glucosidase) [43,44] and, consequently, are potentially bioaccessible and in condition to promote their AA [45]. 

The results showed the highest values of hydroxycinnamic acids and hydroxybenzoic acids in OH extracts than in CONV (Table 1 and Table 2). These compounds frequently exist as bound phenolics in the form of glycosides, amides, esters, and (rarely in free form) may be released by enzymes action that could break these linkages [44]. An example of hydroxycinnamic acid is the chlorogenic acid, which results from the linkage of caffeic acid and quinic acids [46]. Thus, its bond is broken by gastric enzymes, and the free hydroxycinnamic acids are better released throughout GID, increasing the AA [46]. Table 2 confirms the increase of caffeic acid and hydroxybenzoic acid during the digestion process, which are rapidly absorbed in the stomach, and small intestine36 and consequently increase the antioxidant capacity, as Figure 4 confirms. Also, these soluble phenolic acids reduce the hydroxyl group [46]. The AA also depends on the carotenoids released throughout the GID, and its hydrophobicity differences could explain the recovery index differences found in the OH and CONV samples. The CONV extract is hydrophobic, which allows the carotenoids to dissolve in dietary lipids and by bile acids and enzymes lead to its incorporation into micelles and solubilization in the system [43].

The differences between ORAC and ABTS are shown in Figure 4. In ORAC, the AA of OH extract is slightly higher than when measured by ABTS. Unlike the ABTS, which measures the antioxidant activity and the specific reducing power, the ORAC allows to assess the antioxidant scavenging activity and determines the antioxidant status in biological systems [47].

In each phase of digestion, the OH presented higher values than the CONV extracts (*p* < 0.05) for both methods, associated with the higher release of polyphenols.

The tomato extracts are rich in bioactive compounds, including carotenoids and phenolics, released throughout digestion and increase the AA [5,31,42]. The AA of OH extracts increased during digestion. This increase is supported by the direct correlation between AA measured by ABTS and total phenolic compounds, carotenoids and individual compounds r^2^ > 0.84. Besides, the increase of AA may be derived from the formation of new bioactive compounds with antioxidant properties, which may consist of bioactive metabolites generated from modifications of compounds, broken bounds, and others that may originate from metabolic reactions [36]. The increasing recovery index confirms this throughout the GID. The OH samples presented a recovery of 130% index in the mouth, and an AA increase between undigested and digested OH samples was also shown. In the small intestine, the recovery index of OH increased up to 160%, which confirms the saliva, gastric enzymes action, and the bile salts effects on compounds and, thus, its better release throughout the GID system, confirmed by the antioxidant increased. According to the results, the OH presented lower bioaccessibility than the CONV samples (51% and 89%, respectively). 

As explained above, the compounds present in the matrix are digested differently [43,46], translating into the compounds’ bioaccessibility. In this case, the compounds are mainly phenolic compounds and carotenoids and confer antioxidant activity throughout the GID [43].

Both digested extracts, OH and CONV, showed a significant decrease in AA in ORAC and ABTS in the colon (large intestine) and a statistical increase in the basolateral fraction. The results suggest the solubilization of the compound during the GID and diffusion out of the dialysis tube, being bioaccessible [2], reducing the level maintained in the colon.

#### 3.3.2. Prebiotic Effect of Digested OH and CONV Extracts

The effect of tomato extracts after GID simulation, representing the fraction collected in the colon, on the growth of the potential prebiotics can be seen in Figure 5. 

*Bifidobacterium* and *Lactobacillus*, found in the human gastrointestinal microbiota, play an essential role in health promotion. Considering bacterial potential beneficial effects, they are often used in probiotic preparations or considered as target microorganisms for prebiotic substrates [48]. 

The in vitro evaluation of the impact of the tomato bagasse extracts (OH and CONV), after GID passage, upon Bifidobacterium strains has shown two different behaviors. Both bacteria growth as measured by turbidity at 660 nm under the effect of both digested extracts tested and the fructooligosaccharides (FOS) (used as control) at the same concentrations were tested and presented in Table 2.

For *B. animalis* subsp. *lactis* BB-12 (Figure 5A), the OH extracts promoted more minor growth increment (growth rate in the exponential phase) than the CONV extracts (*p* > 0.05). But that is without significant difference for the maximum growth rate reached. Both extracts appeared to promote the growth of the microorganisms, with total biomass after 24 h incubation being higher than the negative control and even with better performance than FOS. Concerning *B. longum* (Figure 5B), the OH also presented a lower growth rate than the CONV samples. Furthermore, the OH showed higher absorbance (biomass) than the CONV samples (*p* < 0.05) and presented similar results to the positive control. To *B. animalis* (Figure 5C), there is an initial lag phase for all of the samples (particularly FOS, which maintains for 20 h). Also, the presence of both extracts had no impact on this bifidobacterial growth comparatively with the positive control (FOS), presenting similar results to the negative control (bacteria growth). As established by some studies, different Bifidobacterium strains may have different carbohydrate utilizing abilities [48,49]. 

Regarding *L. acidophilus* the OH extracts promoted slightly more growth than CONV samples. Also, it presented higher absorbance than the CONV extracts (Figure 5D). Nevertheless, although the OH presented higher absorbance at 6 h when compared with the negative control, after 15 h, the extracts displayed similar results (*p* > 0.05). For *L. casei* the OH sample promoted a faster growth (higher growth rate during 24 h) than CONV one and negative control (Figure 5E). The OH presented higher absorbance in the stationary phase than the CONV extracts (*p* < 0.05). For FOS, there is a prolonged lag phase, but after 24 h, it induced recovery, promoting higher maximum growth than the other samples. For *L. rhamnosus* the OH and CONV samples presented a slower growth rate than the positive control (Figure 5F)). Also, OH extract showed a higher growth rate than the CONV one. These results follow the results presented before, where the OH possesses more glucosides compounds that can promote microorganism growth [49]. OH presented lower absorbance (biomass) compared to the positive control (*p* < 0.05), but higher than the negative control and especially than the CONV extract. The results indicate that this probiotic bacterium may use tomato extracts as a potential prebiotic.

These results indicated that OH extracts promote different probiotic strains’ growth, probably due to the presence of monosaccharides, polyphenols and carotenoids. Gullón et al., (2015) studied the arabinoxylooligosaccharides from wheat bran, and they found different probiotic growth profiles to the same extract. Thus, the extract impact is strain-dependent, as mentioned in previous works in the literature [33,49].

Costa et al., (2019) observed grape flour’s prebiotic and antimicrobial effect, rich in xylooligosaccharides. The authors have also shown that extracts stimulate the growth of probiotics differently. On the other hand, the same extract has selective antimicrobial effects for Gram-positive and negative bacteria. This result reinforces the notion that those extracts present a selective capacity to stimulate bacteria. L. casei, *B. longum* BG3 and B. animalis subsp. OH extracts enhanced *B. animalis* subsp. *lactis* BB-12 growth.

Furthermore, this extract promoted a faster growth rate for *L. casei* and *B. animalis* subsp. *lactis* BB-12 than the positive control. The extracts did not affect the *L. acidophilus*, *L. rhamnosus* and *B. animalis* B0 growth.

#### 3.3.3. Inhibition of Angiotensin-Converting Enzyme Activity (iACE)

Angiotensin-converting enzyme (ACE) is one of the leading regulators of blood pressure. Some anti-hypertensive agents act through the suppression of the enzyme. The in vitro test that assessed ACE inhibitory action can determine the potential anti-hypertensive activity. The basolateral fraction (corresponding to the absorbed samples) evaluated the ACE inhibition since this fraction will be distributed through the bloodstream. 

Both OH and CONV tomato bagasse extracts inhibited ACE (Figure 6A). Statistical results indicated no significant difference in the inhibitory action between both extracts. These extracts contain polyphenols, mainly rutin, with a positive correlation to IACE, r^2^ < 0.98. Several studies have confirmed the rutin inhibitory potential against the ACE activity by its capacity to bind to active sites of the enzyme, competing with the substrate (Ang I) [50,51]. The ACE inhibitory capacity of flavonoids seems to depend on their total flavan-3-ol content, depending on the mean degree of polymerization (mDP) of proanthocyanins, since the monomers such as epicatechin and catechin do not inhibit. Besides, as ACE is a membrane protein, the capacity of procyanidins to be adsorbed on the membrane surface is dependent on the hydroxyl groups on the procyanidins [52].

Other authors found IC50 values of 1.5 mg/mL ACE-inhibitory activities for tomato processing by-products [50]. Nevertheless, they only studied the total fraction of tomato, with no result to the digested fraction and, consequently, the absorbed fraction.

#### 3.3.4. Anti-Inflammatory Inhibition

The results show (Figure 6B,C) inhibition of the release of pro-inflammatory cytokines by peripheral blood mononuclear cells (B and T lymphocytes and monocytes) from healthy donors.

Acute inflammation is a short-term activity, usually appearing within a couple of minutes or times and stopping upon removing the harmful stimulus. It requires the coordinated and general mobilization response of several immunes, endocrine, and neurological mediators of the acute-phase inflammation reaction [18,53] The typical good reaction turns into activated, clears the pathogen, starts the repair process, and then ceases. Interleukine- 6 (Il-6) and tumor necrosis factor-alpha (TNF-α) are examples of cytokines involved in systemic inflammation [53]. The OH and CONV samples’ anti-inflammatory activity was assessed by measuring their ability to reduce the level of Il-6 and TNF-α (tumour necrosis factor) after stimulation with a mixture of lipopolysaccharides (LPS), as shown the Figure 6. The positive control results (cells stimulated only with LPS) are also included in the figure. In vitro results strongly suggest that OH and CONV inhibited LPS-stimulated pro-inflammatory cytokines Il-6 and TNF-α. Both absorbed fractions, OH and CONV, decreased the Il-6 (*p* < 0.05) and TNF-α concentrations (*p* > 0.05). These results could be explained mainly by compounds absorbed in the blood, such as carotenoids (lycopene), phenolics (naringenin, quercetin, rutin), which presented anti-inflammatory properties [18,53]. This bioaccessibility of ingredients is exceptionally contingent on the composition of the nutrients array. Indeed, studies, up to now, suggest a process impact on the bioaccessibility of ingredients that is contingent on this type of compound, the structure and arrangement of the food matrix and the extraction solvents method used. The solubility of bioactive compounds in the water can be separated into lipophilic and hydrophilic. 

The results showed that the CONV samples contain more carotenoids and flavonoids than the OH samples, presenting a slightly higher anti-inflammatory activity. Also, the samples could be encapsulated to withstand the stomach acidity and degradation by enzymes to enhance its absorption. These results follow others reported with literature, whereas authors attributed the anti-inflammatory activity mainly to lycopene from the tomato juice. Riso et al. [54], exhibited modest effects on the production of TNF-α, due to tomato drink by young, healthy volunteers. Mohri et al. [55] study the anti-inflammatory activity of different compounds extracted from tomatoes on obese people. They concluded that tomatoes containing diverse anti-inflammatory compounds could help respond to the chronic inflammation in obese adipose tissue.

In conclusion, the OH extracts from tomato by-products likewise demonstrated anti-inflammatory activity, antioxidant, prebiotic activity stain dependent, and moderate anti-hypertensive action.

## 4. Conclusions

The bioactive compounds extracted from tomato bagasse extracts using a green (OH) and a CONV technology were studied regarding their stability throughout the GIDtract, demonstrating the predicted bioaccessibility of the main compounds present–carotenoids and polyphenols. Both extracts are rich in carotenoids, mainly lutein, lycopene, and β-carotene, as well as polyphenols, mainly hydroxycinnamic acids, and benzoic acids. Although the CONV extract is richer in polyphenols, the OH extract presented a higher TPC recovery index throughout the digestion process, reaching similar concentrations to CONV only in the small intestine. The main phenolic compounds contributing to OH’s higher TPC recovery index are 4-hydroxybenzoic acid, rutin, naringenin, and luteolin. Regarding the total carotenoid content, the content is slightly higher in OH extract, but very similar recovery indexes throughout the GID were observed for both extracts, with a significant decrease after the gastric step. In any case, a higher bioaccessibility was observed for carotenoids present in CONV extract than for OH extract.

Furthermore, the changes caused throughout the digestion process on bioactive compounds also contributed to their bioactivities. Regarding AA, the OH bioaccessibility is also higher than in CONV samples, similar to the trend observed for polyphenols. Furthermore, the OH extracts showed a potential prebiotic effect on *B. animalis*, *B. longum* BG3, and *L. casei*.

Both extracts also presented anti-hypertensive activity in their bioaccessible fraction, although OH bioaccessible compounds showed slightly higher ACE inhibitory capacity (32%) than the CONV ones (28%).

In conclusion, the type of extraction showed to profoundly influence the bioaccessibility and carotenoids of tomato bagasse extracts and related biological properties. The OH extraction, a green alternative technology, improved the bioaccessibility of polyphenols compared to CONV and improved the antioxidant bioaccessibility. Furthermore, OH extract showed a potential prebiotic effect, anti-hypertensive activity, and anti-inflammatory activity with better performance than CONV.

## Figures and Tables

**Figure 1 foods-11-01064-f001:**
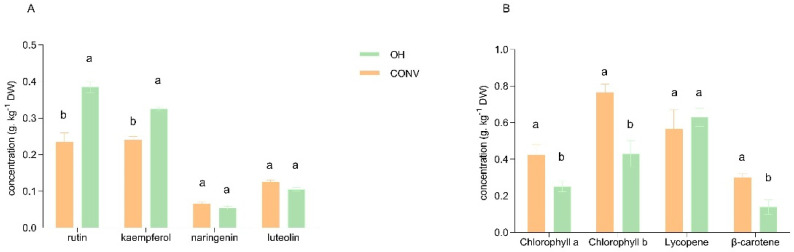
Concentration of (**A**) principal phenols (**B**) chlorophylls and carotenoids present in CONV and OH extracts. Letters represents the significant differences between compound recovery from OH and CONV method used, *p* < 0.05.

**Figure 2 foods-11-01064-f002:**
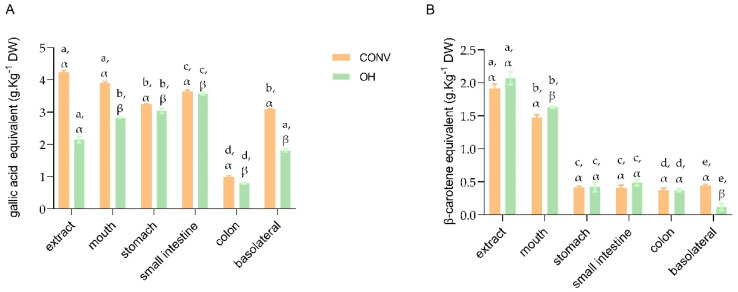
TPC (**A**) and TC (**B**) during the digestion process. TPC (**A**) and total carotenoids (**B**) assays. Significant differences (*p* < 0.05) concerning the gastrointestinal impact for each extract are indicated by Latin letters. Significant differences (*p* < 0.05) among extracts for each digested fraction are indicated by Greek letters.

**Figure 3 foods-11-01064-f003:**
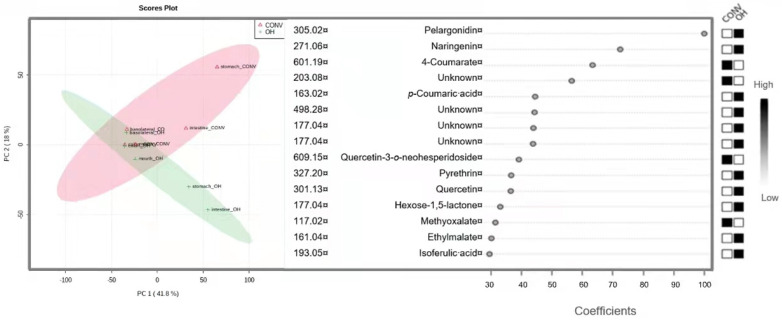
PCA, compounds with significant impact by co-expression analysis and compounds impact on OH and CONV extracts by PLS-DA analysis.

**Figure 4 foods-11-01064-f004:**
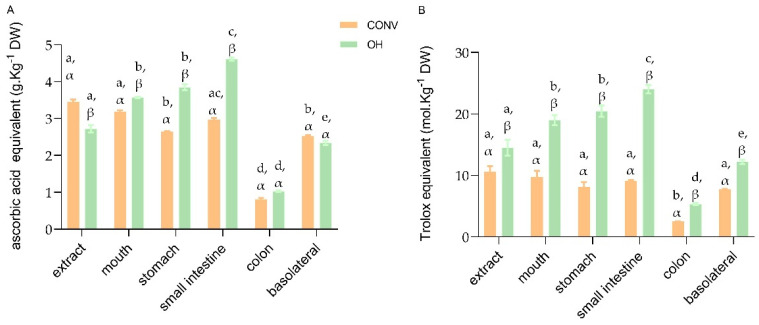
Total antioxidant capacities were determined using the ABTS (**A**) and ORAC (**B**) assays of the in vitro digested fractions derived from tomato liquid extracts (obtained by OH and CONV extraction). Significant differences (*p* < 0.05) among gastrointestinal for each extract are indicated by Roman letters. Significant differences (*p* < 0.05) among extracts for each digested fraction are indicated by Greek letters.

**Figure 5 foods-11-01064-f005:**
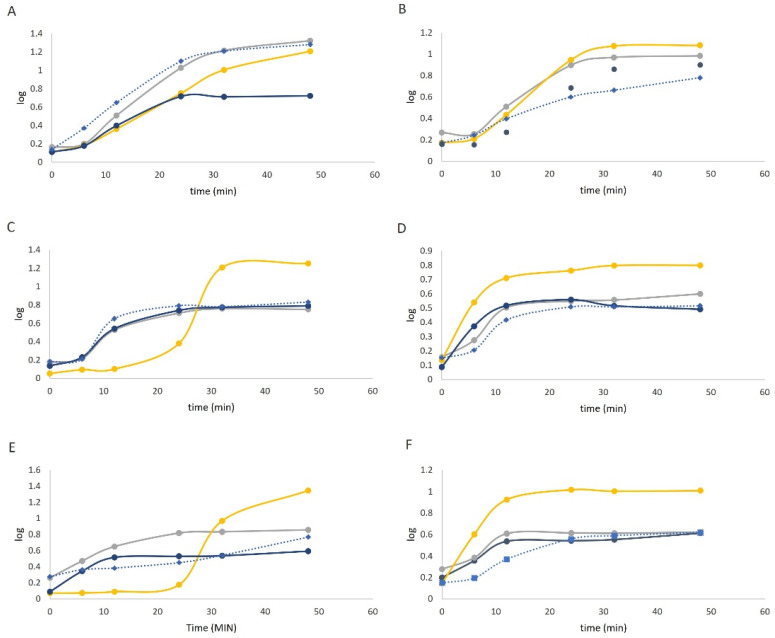
Growth curves of (**A**)—*B. animalis* subsp. *lactis* BB-12; (**B**)—*B. longum* BG3; (**C**)—*B. animalis* Bo; (**D**)— *L. acidophilus* LH5; (**E**)—*L. casei* LC1; (**F**)—*L. rhamnosus* R11 in the presence of 1% extract obtained by OH process (
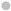
), 1% extract obtained with CONV process (
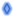
), positive control—FOS (
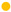
) and negative control without sugar source (
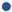
).

**Figure 6 foods-11-01064-f006:**
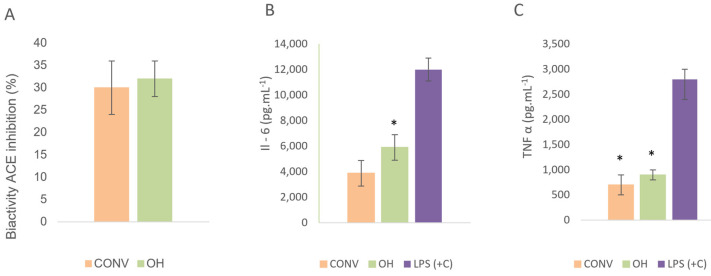
Bioactivity ACE inhibition (%), graph (**A**); Interleukin-6 (Il-6), graph (**B**), tumor necrosis factor-alpha (TNF-α), graph (**C**), concentration in the supernatants from cells stimulated with an absorbed fraction of OH and CONV extracts in combination with 2.5 µg/mL lipopolysaccharide (LPS) per well. The data are expressed as the mean ± SD (* *p* < 0.05 vs. LPS control).

**Table 1 foods-11-01064-t001:** UPLC-qTOF-MS compounds profile of OH and CONV phenolic liquid extracts from tomato bagasse.

Name	OH Samples	CONV Samples	tr (min)	DAD	Formula	*m*/*z* Experimental	*m*/*z* Calculated	MSMS Fragments					Err [mDa]
No identified	✓	✓	7.7	295	C_15_H_2_ON_2_O_4_	291.1355	291.1350	291.1 (22)	171.1 (51)	145.0 (51)	119.1 (100)	145.0 (19)	0.5
Caffeoyl-glucose	✓	✓	8.3	370	C_15_H_18_O_9_	341.0897	341.0878	341.1 (6)	179.0 (100)				1.9
Quinic acid	✓	✓	8.5	330	C_7_H_12_O_6_	191.6634	192.0634	191.0 (100)	146.9 (33)	119.0 (54)	102.9 (75)		0.4
Kaempferol 3-sophorotrioside	✓	✓	8.7	266/360	C_33_H_40_O_21_	771.2004	771.1989	771.2 (100)	609.1 (25)	463.1 (6)	301.0 (4)		1.4
(2-phenylethyl) alpha-L-gulo-hexopyranosyl-(1->6)-alpha-L-gulo-hexopyranoside	✓	✓	9.8	285	C_20_H_30_O_11_	445.1709	445.1715	445.1 (12)	267.1 (31)	221.1 (28)	179.0 (100)		0.6
Quercetin-3-O-sophoroside	✓	✓	10.5	266/366	C_27_H_30_O_17_	625.1421	625.1410	300.0 (74)	179.0 (6)	741.2 (100)			1.1
Quercetin 3-(2G-xylosylrutinoside)	✓	✓	10.9	266/356	C_32_H_38_O_20_	741.0956	741.19564	714.2 (0.01)	300.0 (22)				1.1
Malvidin 3-(6-malonylglucoside) 5-glucoside	✓	✕	11.1	510	C_32_H_38_O20	741.1895	741.884	741.14 (100)	300.0 (16)				0.7
Quercetin-3-*O*-neohesperidoside	✓	✓	11.3	262	C_27_H_30_O_16_	609.1474	609.1461	609.1 (100)	429.1 (3)	284.0 (53)	179.0 (4)		1.3
*p*-coumaric acid	✓	✓	11.3	314	C_9_H_8_O_3_	163.04	163.0401	163.0 (14)	119.0 (100)				0.1
Rutin	✓	✓	11.8	262/356	C_27_H_30_O_16_	609.1442	609.1461	609.1 (100)	301.0 (34)	300.0 (38)	179.0 (2)	151.0 (1)	1.9
Phloridzinyl glucoside	✓	✓	12.2	290	C_27_H_34_O_15_	597.1825	597.1824	597.2 (100)	477.1 (33)	417.1 (24)	387.1 (76)	357.1 (100)	0.1
N-acetyl-D-tryptophan	✓	✓	12.4	281	C_13_H_14_N_2_O_3_	245.0936	245.0932	245.1 (53)	203.1 (100)	159.1 (5)	116.0 (29)	98.0 (7)	0.5
Naringenin-7-*O*-glucoside	✓	✓	13.7	290	C_21_H_22_O_10_	433.1	433.114	433.1 (2)	271.1 (100)	151.0 (8)			0.5
cis-2-Coumarate	✓	✕	16.2		C_9_H_8_O_4_	163.0396	163.0396	163 (100)					0.5
Flavimycin A	✓	✕	17.3		C_18_H_18_O_9_	377.0862	377.0862	377.1 (56)	341 (100)	215 (33)	179 (56)	161 (11)	1.6
Naringenin	✓	✓	18	290	C_15_H_12_O_5_	271.0608	271.0612	271.1 (84)	177.01 (13)	151.0 (100)	119.0 (32)		0.4
4-Coumarate	✕	✓	11.5		C_9_H8O_3_	163.0401	163.0406						0.5
No identified	✕	✓	16.2		C_54_H_67_O_10_	875.4739	875.4723	874.5 (33.0)	97.0 (100)				1.6
(2S)-2-[[2-(diethylamino)-5-[ethyl(piperidine-1-carbonyl)amino]pyrimidin-4-yl]amino]-3-[4-(pyrrolidine-1-carbonyloxy)phenyl]propanoic acid	✕	✓	20.7		C_30_H_42_N_7_O_5_	580.3253	580.32474	580.0 (1)	514.3 (100)				0.6

✓ Compounds identified in samples and X–ar3e the compounds not identified in samples.

**Table 2 foods-11-01064-t002:** Recovery index and bioaccessibility of total phenolic compounds, total carotenoids and antioxidant activity by spectrophotometric analysis and individual compounds by HPLC-DAD analysis from CONV and OH extracts throughout digestion.

Compounds	Recovery Index (%)										Bioaccesibility Index (%)	
	Mouth		Stomach		Small Intestine		Colon		Basolateral			
	CONV	OH	CONV	OH	CONV	OH	CONV	OH	CONV	OH	CONV	OH
Total Phenolic compounds	92.32 ± 1.78 a	131.05 ± 1.25 a	76.53 ± 2.23 b	141.19 ± 7.25 b	86.12 ± 2.14 a	165.98 ± 3.51 c	23.44 ± 1.25 d	36.87 ± 1.78 d	73.10 ± 2.62 b	84.25 ± 1.89 e	75.72 ± 1.45 α	69.56 ± 1.53 β
4-Hydroxybenzoic acid	10.69 ± 1.25 a	94.46 ± 1.84 a	1.35 ± 0.09 b	105.94 ± 3.28 b	2.50 ± 0.56 b	44.00 ± 1.98 c	3.95 ± 1.25 c	9.54 ± 1.33 d	8.18 ± 1.44 d	4.94 ± 0.78 e	67.41 ± 2.73 α	34.13 ± 1.23 β
trans-cinnamic acid	92.19 ± 4.13 a	122.70 ± 5.67 a	n.d.	296.89 ± 8.23 b	n.d.	197.20 ± 3.63 b	n.d.	n.d.	n.d.	n.d.	n.d.	n.d.
vanillin	n.d.	65.51 ± 2.59 a	n.d.	175.37 ± 4.17 b	n.d.	59.83 ± 1.88 a	n.d.	n.d.	n.d.	n.d.	n.d.	n.d.
caffeic acid	96.54 ± 3.32 a	138.16 ± 1.61 a	n.d.	n.d.	n.d.	n.d.	n.d.	n.d.	n.d.	47.73 ± 9.85 b	n.d.	10.00 ± 1.23 α
luteolin	n.d.	167.40 ± 4.21 a	n.d.	143.45 ± 3.78 b	n.d.	57.91 ± 3.21 c	n.d.	101.99 ± 4.45 d	n.d.	n.d.	n.d.	n.d.
rutin	59.71 ± 2.24 a	145.44 ± 2.13 a	3.40 ± 0.78 b	236.33 ± 5.45 b	41.90 ± 2.32 c	131.40 ± 2.41 a	35.72 ± 1.69 d	32.99 ± 2.44 c	33.01 ± 1.12 d	13.72 ± 2.56 d	48.03 ± 3.24 β	29.37 ± 4.71 α
naringenin	n.d.	107.49 ± 3.27 a	n.d.	149.67 ± 2.98 b	n.d.	224.15 ±7.42 c	62.22 ± 4.44 a	67.60 ± 1.94 d	52.50 ± 2.41 a	12.95 ± 1.76 e	45.76 ± 1.85 β	16.08 ± 1.89 α
Total carotenoids	77.03 ± 4.67 a/α	78.75 ± 2.25 a/α	21.58 ± 2.85 a/β	20.65 ± 2.43 a/β	21.27 ± 4.25 a/β	23.67 ± 2.81 a/β	19.38 ± 0.97 a/β	17.92 ± 1.71 a/β	22.967 ± 2.43 a/β	5.76 ± 3.12 a/ỿ	93.62 ± 1.41 α	25.86 ± 1.12 β
Total antioxidant activity	92.75 ± 2.01	104.08 ± 12.99	76.23 ± 4.67	115.51 ± 12.01	86.38 ± 3.21	106.53 ± 11.84	23.19 ± 1.24	41.80 ± 2.78	73.04 ± 5.87	95.51 ± 3.32	84.89 ± 2.98 α	89.66 ± 2.44 β

n.d.—not detected; Results are the average of three determinations ± standard deviation. Significant differences (*p* < 0.05) between digestive fractions (mouth, stomach, small intestine, colon nad basolateral) for each extraction method are indicated by greek letters. Significant differences between the extraction methods (OH nad COMV) for each digestive fraction are indicated by roman letters, as determined by one-way ANOVA test (*p* < 0.05), respectively.

## Data Availability

Data are available on request.

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
