# Peer review of "Bioactivity and Bioaccessibility of Bioactive Compounds in Gastrointestinal Digestion of Tomato Bagasse Extracts"

_foods, 2022, doi:10.3390/foods11071064_

Round 1
Reviewer 1 Report
Generally, the paper have good experimental work but although the paper is well structured and the methodology partially developed in a correct way, the paper cannot be considered for publication in its actual format. Below some suggestions:
In vitro always in Italics
Lines 41-44: I suggest describing briefly ohmic heating
Line 60: CONV means conventional? Please the first time write it in full, then you can use it abbreviated
Line 61: gastrointestinal (GI)
Line 64: gastrointestinal digestion (GID)
Lines 84-85: who produces this bagasse? Is it the waste of some company? Or did you create it in the lab? Please specify
Lines 125-127: please check the abbreviations as they are not the same as in the formula above
Lines 130-131: how are total polyphenols expressed? Was the assay carried out in triplicate? Please add
Lines 139-151: please add HPLC elution method as you have done for carotenoids in the next paragraph
Lines 178-179: how were ORAC and ABTS results expressed? Were the assays performed in triplicate? Please add
Line 181: four? Please check
Lines 196-197: please describe this assay briefly. I already have to download other papers to understand how other assays were performed previously. It is difficult and tiring to follow your experimental phase if I have to download too many papers
Lines 201-205: is this a method developed in your lab? If not, please add references
Lines 240-242: please add references
Lines 248: AA? Where is reported the antioxidant activity? In which figure or table? Since you talk much later (in the paragraph 3.3) about the antioxidant activity, I suggest you remove this consideration from here because it only creates confusion
Lines 253-255: please add references
Paragraph 3.1.2: This paragraph is a bit confusing. First you start discussing Figure 1b, then you refer to Figure 2 (lines 296-299), then go back to making considerations on Figure 1 (from lines 299 onwards). You have to rewrite the paragraph avoiding this confusion.
Lines 303-304: please add references
Line 368-384: please add references
Lines 425-430: r2 not r2
Lines 442-423: add reference to Figure 3
Figure 3: make this figure clearer, by increasing the resolution
Figure 5: the legend is unclear
Figure 6: make this figure clearer, by increasing the resolution
Author Response
Dear Reviewer,
First, the authors sincerely acknowledge the interest demonstrated in our work and the availability to reconsider a revised version of this manuscript.
We want to thank all the positive inputs and suggestions given by the reviewer, which contribute to improving and enriching this manuscript.
The answers are given just after the transcription of the reviewer’ comments, and new information is added to the article with tracked changes as requested in the revised version.
REVIEWER REPORT(S):
Reviewer 1
Generally, the paper have good experimental work but although the paper is well structured and the methodology partially developed in a correct way, the paper cannot be considered for publication in its actual format. Below some suggestions:
Q. In vitro always in Italics
R. The expression “in vitro” was improved as suggested.
Q. Lines 41-44: I suggest describing briefly ohmic heating
R. The introduction was improved, please see page 2, lines 45 to 51.
Q. Line 60: CONV means conventional? Please the first time write it in full, then you can use it abbreviated.
R. The expression was improved as suggested, please see page 2, line 51.
Q. Line 61: gastrointestinal (GI).
R. The expression was improved as suggested, page 2, lines 70 to 71.
Q. Line 64: gastrointestinal digestion (GID)
R. The expression was improved as suggested, page 2, line 71.
Q. Lines 84-85: who produces this bagasse? Is it the waste of some company? Or did you create it in the lab? Please specify
R. The bagasses were produced in a tomato processing company. The specification was improved in the document, please see page 3, lines 94 and 95.
Q.Lines 125-127: please check the abbreviations as they are not the same as in the formula above
R. The abbreviations were improved accordingly, please see page 4, lines 137 to 141.
Q. Lines 130-131: how are total polyphenols expressed? Was the assay carried out in triplicate? Please add
R. The method was improved, please see page 4, lines 146 to 149.
Q. Lines 139-151: please add HPLC elution method as you have done for carotenoids in the next paragraph
R. The description was improved accordingly suggestions, please see page 4, lines 165 to 167.
Q.Lines 178-179: how were ORAC and ABTS results expressed? Were the assays performed in triplicate? Please add
R. The description was improved accordingly suggestions, page 5, lines 198 to 215,
Q. Line 181: four? Please check
R. The sentence was improved accordingly, six microorganisms were used.
Q. Lines 196-197: please describe this assay briefly. I already have to download other papers to understand how other assays were performed previously. It is difficult and tiring to follow your experimental phase if I have to download too many papers
R. The assay was described as suggested, please see page 6, lines 232 to 241.
Q. Lines 201-205: is this a method developed in your lab? If not, please add references
R. The method was performed in our lab, the references was added.
Q. Lines 240-242: please add references
R. The references was added, please see line 245.
Q. Lines 248: AA? Where is reported the antioxidant activity? In which figure or table? Since you talk much later (in the paragraph 3.3) about the antioxidant activity, I suggest you remove this consideration from here because it only creates confusion.
R. The sentence was removed.
Q. Lines 253-255: please add references
R. The references were added, page 6, line 387 to 394.
Q. Paragraph 3.1.2: This paragraph is a bit confusing. First you start discussing Figure 1b, then you refer to Figure 2 (lines 296-299), then go back to making considerations on Figure 1 (from lines 299 onwards). You have to rewrite the paragraph avoiding this confusion.
R. The paragraph was rewritten.
Q. Lines 303-304: please add references
R. The references were added.
Q. Line 368-384: please add references
R. R. The references were added.
Q. Lines 425-430: r2 not r2
R. The sentence was modified, lines 494 to 498.
Q. Lines 442-423: add reference to Figure 3
R. The reference to figure 3 was added, please see line 509.
Q. Figure 3: make this figure clearer, by increasing the resolution
R. The figure was improved.
Q. Figure 5: the legend is unclear
R. The legend was improved.
Q. Figure 6: make this figure clearer, by increasing the resolution
R. The figure was improved

Reviewer 2 Report
In this paper the authors investigated in tomato bagasse the effect of the application of ohmic heating technology followed by an extraction with hydroalcoholic solution at different temperatures (70 ° C and 55 ° C) compared to an extraction with ethanol or hexane without being preceded from ohmic heating on the recovery of phenolic compounds and carotenoids. Subsequently, the lyophilised extract underwent in vitro digestion and the bioaccessibility and bioactivity of bioactive compounds was assessed. The article is poorly written, difficult to read, the methods are incomplete and report descriptive errors. In addition, the design of the study is incorrect.
Major revision:
It makes no sense to compare the effect of a technology such as ohmic heating when the extraction methods are different (hydroalcoholic extraction at 70 and 55°C preceded by ohmic heating vs extraction with ethanol and hexane not from ohmic heating).
In vitro digestion is a useful tool to evaluate the role of the matrix effect on nutrient bioaccessibility. In this study, a freeze-dried extract was digested, which therefore does not have a food matrix effect, in addition to the fact that it is not commonly taken with the diet as it is. Therefore the applied experimental conditions do not find a parallel with reality.
The article has numerous inaccuracies and errors that make one seriously doubt the quality and attention that may have been applied during the experimental part in data acquisition.
Editing errors such as no spaces between characters (line 33, 36, 40, 224...), excessive line spacing, double spaces...
Misuse of the terms bioacessibility and bioavailability without indicating the true meaning of the term. Bioaccessibility is the release of nutrients of the food matrix during digestion with respect to their content in the food and not that indicated in line 121. In addition to the fact that the dialysis step is not sufficient to evaluate bioavailability because it does not take into account the role of membrane transporters.
Failure to define acronyms (lines 112, 118, 124..)
Failure to explain abbreviations. What do the authors mean by IN and OUT? (line 126, 127)
In the description of in vitro digestion the authors do not report essential information such as the concentration of enzymes, the porosity of the membranes, the reference from which the protocol was used...
The ORAC method is not described in the methods
Gallic acid is not used as a calibration line for the determination of the total carotenoid content, but in the Folin assay
The antioxidant activity evaluated with the abts assay is not a biological property (line 174)
Line 181. You need to enter a reference
Line 196. On what was the ACE inhibitor activity evaluated? iACE is not the correct acronym
Figure 1. The statistical symbols in the figure do not correspond with what is reported in the legend. The presence of letters in the figure presupposes the post test of Tukey but it is clear that the comparison has not been made all counted as predicted by the Tukey but only but only between the two experimental conditions for each bioactive. * not present in figure
Author Response
Dear Reviewer,
First, the authors sincerely acknowledge the interest demonstrated in our work and the availability to reconsider a revised version of this manuscript.
We want to thank all the positive inputs and suggestions given by the reviewer, which contribute to improving and enriching this manuscript.
The answers are given just after the transcription of the reviewer’ comments, and new information is added to the article with tracked changes as requested in the revised version.
REVIEWER REPORT(S):
Reviewer 2
In this paper the authors investigated in tomato bagasse the effect of the application of ohmic heating technology followed by an extraction with hydroalcoholic solution at different temperatures (70 ° C and 55 ° C) compared to an extraction with ethanol or hexane without being preceded from ohmic heating on the recovery of phenolic compounds and carotenoids. Subsequently, the lyophilised extract underwent in vitro digestion and the bioaccessibility and bioactivity of bioactive compounds was assessed. The article is poorly written, difficult to read, the methods are incomplete and report descriptive errors. In addition, the design of the study is incorrect.
Major revision:
Q. It makes no sense to compare the effect of a technology such as ohmic heating when the extraction methods are different (hydroalcoholic extraction at 70 and 55°C preceded by ohmic heating vs extraction with ethanol and hexane not from ohmic heating).
R. Thanks for the suggestions. In fact, there is a communication error. the objective of this study was not to compare the extraction methods as technology.
It was intended to understand if bioactive extracts (rich in phenolic compounds and carotenes produced by two different methods - ohmic treatment (under optimal conditions) and conventionally method (results already published - [1–3]) will impact the bioaccessibility tof bioactives throughout gastrointestinal tract. We emphasize that the extraction yield of the compounds in question is not relevant in this work. In previous works, the ohmic was evaluated under the same conditions as the conventional one (solvent, temperature and extraction time) to verify the effects of the ohmic treatment.
Q. In vitro digestion is a useful tool to evaluate the role of the matrix effect on nutrient bioaccessibility. In this study, a freeze-dried extract was digested, which therefore does not have a food matrix effect, in addition to the fact that it is not commonly taken with the diet as it is. Therefore the applied experimental conditions do not find a parallel with reality.
R. yes, you're absolutely right, but it could be used as a functional ingredient and that before be incorporated in a model food matrix needs to me tested to understand how the compounds behave throughout the tract and if the nature of the treatment may affect the bioacessibility of the key bioactive compounds of the ingredient and later a prove of concept in food matrices are expected to be done.
The article has numerous inaccuracies and errors that make one seriously doubt the quality and attention that may have been applied during the experimental part in data acquisition.
Q. Editing errors such as no spaces between characters (line 33, 36, 40, 224...), excessive line spacing, double spaces...
R. The improvement was done, as suggested.
Q. Failure to define acronyms (lines 112, 118, 124..)
R. The improvement was done as suggested, please see line 70, 124,138.
Q. Failure to explain abbreviations. What do the authors mean by IN and OUT? (line 126, 127)
R. The improvement was done, please see lines 120 and 139 to 141.
Q. In the description of in vitro digestion the authors do not report essential information such as the concentration of enzymes, the porosity of the membranes, the reference from which the protocol was used...
R. The method was improved.
Q. The ORAC method is not described in the methods
R. the ORAC method was described accordingly.
Q. Gallic acid is not used as a calibration line for the determination of the total carotenoid content, but in the Folin assay
R. The total carotenoid content was expressed in gβ-carotene equivalent/Kg. The description was changed accordingly.
Q. The antioxidant activity evaluated with the abts assay is not a biological property (line 174).
R. The description was improved accordingly.
Q. Line 181. You need to enter a reference
R. The reference was improved.
Line 196. On what was the ACE inhibitor activity evaluated? iACE is not the correct acronym
R. The method evaluates the anti-hypertensive activity. The description was improved.
Q, Figure 1. The statistical symbols in the figure do not correspond with what is reported in the legend. The presence of letters in the figure presupposes the post test of Tukey but it is clear that the comparison has not been made all counted as predicted by the Tukey but only but only between the two experimental conditions for each bioactive. * not present in figure
R. The statistical was improved as suggested.
References
1. Coelho, M.; Pereira, R.N.; Teixeira, J.A.; Pintado, M. Valorization of Tomato Wastes: Influence of Ohmic Heating Process on Polyphenols Extraction Time. In Proceedings of the Extended abstract of Vienna Polyphenols 2017; Archives of International Society of Antioxidants in Nutrition.: Vienna, 2017; pp. 37–40.
2. Coelho, M.; Pereira, R.; Rodrigues, A.S.; Teixeira, J.A.; Pintado, M.E. Extraction of Tomato By-Products’ Bioactive Compounds Using Ohmic Technology. Food and Bioproducts Processing 2019, 117, 329–339, doi:10.1016/j.fbp.2019.08.005.
3. Coelho, M.; Silva, S.; Costa, E.; Pereira, R.N.; Rodrigues, A.S.; Teixeira, J.A.; Pintado, M. Anthocyanin Recovery from Grape By-Products by Combining Ohmic Heating with Food-Grade Solvents: Phenolic Composition, Antioxidant, and Antimicrobial Properties. Molecules 2021, 26, doi:10.3390/molecules26133838.

Reviewer 3 Report
The manuscript entitled "Bioactivity and bioaccessibility of bioactive compounds in gastrointestinal digestion of tomato bagasse extracts" requires some minor comments and considerations:
- Please to revise the manuscript according to the recommended guidelines of Foods. An example is the title, each word is capitalized and italics should not be used. The spaces are also missing in many places.
- In the introduction, please include information about the types of polyphenolic compounds found previously in tomatoes. In the introduction, please include information about the types of polyphenolic compounds found in tomatoes. Since the Authors identified a number of little-known compounds in tomato pomace, it is important to show whether they were ingredients already identified in tomatoes earlier.
- n table 1, please clarify the nomenclature of Quercetin 3- (2G-xylosylrutinoside), Phene-di-hexoxe (or Phene-di-hexose?). Please give me the name of the compound: (2S) -2 - [[2- (diethylamino) -5-[ethyl (piperidine-1-carbonyl) amino] pyrimidin-4-yl] amino] -3- [4- (pyrrolidine-1-carbonyloxy) phenyl] propanoic acid; this compound was only identified on the basis of the molecular weight 580.3253, what was its MS fragmentation?
- In Table 1, O (oxygen) is not always in italics.
- According to the Authors, kaempferol compounds 3-sophorotrioside, phene-di-hexoxe (phene-di-hexose ?) and N-acetyl-D-tryptophan were identified for the first time in the species/in the nature. Therefore, please to present their more precise MS identification.
Author Response
Dear Reviewer,
First, the authors sincerely acknowledge the interest demonstrated in our work and the availability to reconsider a revised version of this manuscript.
We want to thank all the positive inputs and suggestions given by the reviewer, which contribute to improving and enriching this manuscript.
The answers are given just after the transcription of the reviewer’ comments, and new information is added to the article with tracked changes as requested in the revised version.
REVIEWER REPORT(S):
The manuscript entitled "Bioactivity and bioaccessibility of bioactive compounds in gastrointestinal digestion of tomato bagasse extracts" requires some minor comments and considerations:
Q. Please to revise the manuscript according to the recommended guidelines of Foods. An example is the title, each word is capitalized and italics should not be used. The spaces are also missing in many places.
R. The suggestions were changed as suggested.
Q. In the introduction, please include information about the types of polyphenolic compounds found previously in tomatoes. Since the Authors identified a number of little-known compounds in tomato pomace, it is important to show whether they were ingredients already identified in tomatoes earlier.
R. The introduction was improved accordingly, please see page 2, lines 62 to 64.
Q. In table 1, please clarify the nomenclature of Quercetin 3- (2G-xylosylrutinoside), Phene-di-hexoxe (or Phene-di-hexose?). Please give me the name of the compound: (2S) -2 - [[2- (diethylamino) -5-[ethyl (piperidine-1-carbonyl) amino] pyrimidin-4-yl] amino] -3- [4- (pyrrolidine-1-carbonyloxy) phenyl] propanoic acid; this compound was only identified on the basis of the molecular weight 580.3253, what was its MS fragmentation?
R. The changes were improved as suggested. Please see table 1. The Quercetin 3-(2G-xylosylrutinoside) is the synonym of Quercetin 3-O-beta-(2(G)-O-beta-xylopyranosyl-6(G)-O-alpha-rhamnopyranosyl)glucopyranoside. Relatively to (2S) -2 - [[2- (diethylamino) -5-[ethyl (piperidine-1-carbonyl) amino] pyrimidin-4-yl] amino] -3- [4- (pyrrolidine-1-carbonyloxy) phenyl] propanoic acid is the name give for pubchem based only by molecular weight.
Q. In Table 1, O (oxygen) is not always in italics.
R. The table was improved accordingly
Q. According to the Authors, kaempferol compounds 3-sophorotrioside, phene-di-hexoxe (phene-di-hexose ?) and N-acetyl-D-tryptophan were identified for the first time in the species/in the nature. Therefore, please to present their more precise MS identification.
R. MS fragmentation was improved accordingly suggestions. These compounds are often used in industry and, as our samples come from tomato processing, there may be cross-contamination (e.g. pesticides and other production chemicals). The method was also improved, please see page 5, lines 187 to 192

Round 2
Reviewer 2 Report
The manuscript continues to have serious conceptual and experimental design gaps that the authors have failed to improve or convincingly explain in the answers. In particular the authors replied: "It was intended to understand if bioactive extracts (rich in phenolic compounds and carotenes produced by two different methods - ohmic treatment (under optimal conditions) and conventionally method (results already published - [1–3]) will impact the bioaccessibility of bioactives throughout gastrointestinal tract. " While as indicated in the manuscript in the section Methods paragraph Samples: "the CONV method with organic solvents, such as ethanol (70%) and hexane, were used to extract phenolic compounds, and carotenoids, respectively [13]. In OH, hydroethanolic solutions of 70% ethanol were used as selected solvent extraction, during 15 min, at 70 ºC and 55 ºC for phenolic and carotenoids compounds.". Clearly, for example in the case of carotenoids, there are various variations in the application of the technology (HO vs CONV) and the type of extraction (hexane vs hydroalcholic at 55°C). This makes it impossible to discriminate the effect of the technology or extraction technique in subsequent results.
Although some authors have modified some sentences and introduced explanations following the suggestions of the reviewers, within the manuscript there are errors that denote a low attention on the part of the authors in the writing phase of the manuscript (eg: summary description in the legends of the figures, the description of the Y axis in figure 2B does not correspond with the carotenoids, formatting and English errors...)
Author Response
Dear Reviewer,
First, the authors sincerely acknowledge the interest demonstrated in our work and the availability to reconsider a revised version of this manuscript.
We want to thank all the suggestions which improved and enriched this manuscript. English has been carefully reviewed, as Figure 2 was improved.
The description of materials and methods has also been improved to make the objective of the work perceptive. The authors are aware that different extracts will have different behaviours along the gastrointestinal tract. These same differences are of interest to verify which bioactivities are obtained and whether the fact that they use different extracts loses or gains biological activities, the metabolites formed, and their bioaccessibility.
The extraction yield is not the study's objective, and this is already discussed and published by the authors in tomato and grape by-products matrices. In addition, we are trying to respond to global challenges for a more sustainable economy, a greener environment, and a circular economy as a solution.
We hope to have been more transparent, and we are available for any questions.
Once again, we appreciate the comments, as they help us to be critical, to do more and better,
Best regards,
Manuela Pintado
